# Idiopathic Pulmonary Hemorrhage in Infancy: A Case Report and Literature Review

**DOI:** 10.3390/diagnostics13071270

**Published:** 2023-03-28

**Authors:** Fabio Del Duca, Aniello Maiese, Federica Spina, Giacomo Visi, Raffaele La Russa, Paola Santoro, Maria Serenella Pignotti, Paola Frati, Vittorio Fineschi

**Affiliations:** 1Department of Anatomical, Histological, Forensic and Orthopedical Sciences, Sapienza University of Rome, Viale Regina Elena 336, 00161 Rome, Italy; 2Department of Surgical, Medical and Molecular Pathology and Critical Care Medicine, Section of Legal Medicine, University of Pisa, 56126 Pisa, Italy; 3Department of Clinical and Experimental Medicine, University of Foggia, 71122 Foggia, Italy; 4Department of Neonatology and Neonatal Intensive Care, Anna Meyer Children’s Hospital, 50139 Florence, Italy

**Keywords:** acute idiopathic pulmonary hemorrhage in infants (AIPHI), sudden infant death syndrome, pulmonary hemorrhage, case report, autopsy, postmortem, forensic, lung histology

## Abstract

Acute idiopathic pulmonary hemorrhage in infants (AIPHI) is a rare and quite low-described entity. Nowadays, pathophysiological mechanisms are poorly understood, although the lethality remains high. We present an autopsy case report of a 2-day-old male who developed respiratory distress and blood leakage from the endotracheal tube (ET) and suddenly died because of acute pulmonary hemorrhage. A postmortem examination and histological analysis were performed and are reported in this paper. Alveolar spaces were filled with red blood cells and hyaline membranes in all the examined samples. The absence of other findings led us to select a post-mortem diagnosis of AIPHI. To support our diagnosis, we conducted a systematic review of the updated scientific literature and found that only 61 cases have been reported. Most of them presented acute respiratory distress and bleeding from the upper airways with blood leakage from ET (9.83%), hemoptysis (52.45%), epistaxis (8.2%), and hematemesis (3.27%). The autopsy data revealed hemorrhages of the lower airways and hemosiderin-laden macrophages. The data from the scientific publications and our findings are essential to achieving a correct diagnosis. On these bases, we suggest autoptic criteria to achieve a post-mortem diagnosis of AIPHI.

## 1. Introduction

Neonatal pulmonary hemorrhage (NPH) can be defined as a massive blood leakage in the terminal sacs and pulmonary interstitium in infants, usually associated with hyaline membrane disease [1,2]. NPH is a catastrophic event that mostly affects preterm infants and can be associated with birth asphyxia [3], hemorrhagic disease in newborns [4], fetal or neonatal pneumonia, ventilation, and surfactant administration [5,6,7,8].

In these cases, a massive lung injury, followed by an imponent loss of blood, precedes infant death. 

However, in some cases, there is no trigger for pulmonary hemorrhage. This condition is described as acute idiopathic pulmonary hemorrhage in infants (AIPHI), a very rare and quite peculiar disease. The criteria for the definition of AIPHI were clearly described by the Centers for Disease Control and Prevention after some clusters occurred in Cleveland, Ohio in the 1990s [9].

AIPHI occurs during the first year of life in full-term, born babies with unremarkable medical histories; these babies suddenly develop severe, acute respiratory distress or respiratory failure due to an evident presence of blood in the airways [10].

Epidemiological data for this distinct clinical presentation are no longer available in the updated scientific literature. In general, pulmonary hemorrhage (PH) commonly occurs within the first few days of life in 1 to 12 per 1000 live births [11], with a mortality rate of 50%. The autoptic data reveal PH incidence to be variable, from 0.86% to 37.3% [11,12,13,14]. Two main factors affect the lack of updated epidemiological data on the disease: its extreme rareness and the absence of diagnostic examinations or post-mortem debriefing of these cases. 

To improve knowledge of this occurrence, we present a rare case of a 3-day-old baby—appearing healthy at birth—who suddenly developed an acute respiratory failure. Subsequently, the autoptic examination confirmed the diagnosis of AIPHI. There was no external evidence of injury that suggested the baby had been abused. This case report aimed to describe a rare case of a lung hemorrhage, without its typical related clinical manifestations, and to explain the method followed in order to achieve the diagnosis. A literature review of sudden idiopathic pulmonary hemorrhage in infants was performed.

## 2. Autopsy Case Report

### 2.1. Clinical Presentation 

This case is about a healthy male newborn who suddenly developed acute respiratory failure and died a few days after delivery. 

The pregnant person was 36 years old and gave birth to him at 38.5 weeks. No pregnancy complications were reported.

Cardiotocography (CTG) was frequently performed during pregnancy and did not reveal any alterations. 

At 38.5 weeks of gestation, a CTG reported mild bradycardia, and the clinical examination revealed a loss of meconium-stained vaginal fluid. An emergency cesarean section (C-section) was successfully conducted at 3:28 a.m. The baby was birthed alive and vigorous and weighed 2820 g. An APGAR evaluation was conducted at minute 1 and minute 5, revealing normal scores (8 at 3:29 and 9 at 3:34 a.m.); consequently, there was no need for resuscitation maneuvers.

A vaginal swab was negative for common pathogens. 

After meconium aspiration, the newborn was transferred to the nursery room for the night. The next morning, he was handed over to his mother for breastfeeding. At 09:30 a.m., he suddenly developed acrocyanosis and polypnea. He was admitted to the intensive care unit, where he was intubated, and imponent blood loss from the ET was detected. Anemia, respiratory failure, and hypotension were treated, but resuscitation maneuvers were useless. Multiple blood tests were performed during hospitalization and no evidence of leukocytosis or high inflammatory indices was detected. The infant died the following day at 10:50 a.m. The parents asked the judicial authority to clarify matters. Then, a forensic evaluation was demanded.

### 2.2. Postmortem Examination 

The autopsy was performed three days after the infant’s death [15]. The total body weight was 2800 g, and the baby was 51 cm tall (61st percentile); both of which were appropriate for his age. Other routine external body measurements were taken (the crown–rump length was 34 cm, the thoracic circumference was 32 cm, the abdominal circumference was 29 cm, the foot length was 7 cm, and the occipitofrontal circumference was 34 cm). 

The external examination did not reveal any abnormalities. The oral mucosa and conjunctiva did not show petechial hemorrhages. The only sign detected was chest compression. 

No gross internal abnormalities were found, and organs appeared normally placed and well developed. Moderate effusion was detected in the pleural and abdominal cavities. No congenital heart malformations were revealed through in situ dissection (heart weight, 30 g). All internal organs were dissected *en block* following the Rokitansky technique. A macroscopic examination was performed before and after formalin fixation. The airways showed hyperemia of the mucous membrane of the larynx and trachea, which were covered with crimson-colored liquid. The lungs (weight left, 55 g; right, 48 g) were purplish, with signs of congestion and salutary hemorrhage infiltration of the pleura. A lung weight-to-body weight ratio (LW/BW) of 0.0195 revealed pulmonary hypoplasia.

### 2.3. Histological Evaluation

The lung specimens showed remarkable findings. The visceral pleura showed increased thickness, with fibrin deposits and both diffuse and nodular hemorrhagic foci. In multiple fields, hemorrhage foci spread into the pulmonary parenchyma. 

In the central zones, there were large, empty spaces among red blood cells, which were derived from extensive and bilateral pulmonary hemorrhage (Figure 1). In all the examined fields, the alveolar lumen was covered with hyaline membranes (Figure 1C,D—arrows). Minimal areas of partial alveolar collapse were found, but there was no presence of evident atelectasis. Only minimally and sporadic spots of meconium were found in the bronchial lumen. No signs of abnormal pulmonary angiogenesis were founded. 

There were no remarkable findings in the other organs except for congestion and cerebral swelling. A histological examination of the placenta (350 gr) revealed no malformations.

An average semi-quantitative evaluation of pulmonary hemorrhage was performed according to the scoring system proposed by Krous et al. 2006 [3]. Microscopic evaluation showed a grade 3 pulmonary hemorrhage.

## 3. Literature Review

### 3.1. Materials and Methods

A systematic review of the updated literature was carried out according to the Preferred Reporting Items for Systematic Review (PRISMA) standards [16].

A systematic literature search and a critical appraisal of the collected studies were conducted. An electronic search of PubMed from the inception of these databases to January 2023 was performed.

The search terms were: (“acute idiopathic pulmonary hemorrhage” OR “Acute pulmonary hemorrhage” OR “acute lung hemorrhage” OR “pulmonary hemorrhage” OR “lung hemorrhage” OR “acute pulmonary haemorrhage” OR “acute lung haemorrhage”) AND (“idiopathic” OR “unexplained” OR “unexpected” OR “sudden”) AND (“infant” OR “newborn” OR “natal” OR “neonatal” OR “fetal”).

The bibliographies of all located papers were examined and cross-referenced to search for further relevant literature. The methodological appraisal of each study was conducted according to the PRISMA standards, including the evaluation of bias. Data collection entailed study selection and data extraction. Two researchers independently examined the papers with titles or abstracts that appeared to be relevant and selected the ones that analyzed clinical presentation, instrumental analysis, and autopsy reports for acute idiopathic pulmonary hemorrhage in infants. Disagreements concerning eligibility between the researchers were resolved using a consensus process. No unpublished, pre-print, or grey literature was searched. Only papers in English were included in the search.

Other exclusion criteria were clinical trials of new drugs, systematic reviews of drugs used in PH treatment, studies that did not fit into the CDC criteria, retrospective studies of overall survival, animal studies, and cases involving no healthy-born children. Data extraction was performed by two investigators and verified by two other investigators. 

This study was exempt from institutional review board approval as it did not involve human subjects.

### 3.2. Results

A review of the titles and abstracts, as well as a manual search of the reference lists, was carried out. The reference lists of all identified articles were reviewed to find missed papers. This search identified 126 articles, which were then screened after reading their abstracts. The resulting 24 reference lists were screened to exclude duplicates and irrelevant papers. In addition, non-English papers were excluded, and the following inclusion criteria were used: (1) original research articles, (2) reviews and mini-reviews, and (3) case reports/series. These publications were carefully evaluated, taking into account the main aims of the review. This evaluation left nine scientific papers comprising original research articles, case reports, and case series. Figure 2 illustrates our search strategy. 

We found articles both on respiratory manifestations and lung pathology in AIPHI. However, a clinical background is fundamental to understanding lung involvement, so a brief description of the respiratory manifestations is provided below. The search strategy is summarized in Figure 2. 

#### 3.2.1. General Presentation and Clinical Manifestation of AIPHI

The spectrum of pulmonary manifestations caused by AIPHI is wide, varying from self-limiting, mild symptoms that may cause hemosiderosis, up to the most severe manifestations, such as hypoxia and respiratory distress.

A total of 61 cases of acute idiopathic pulmonary hemorrhage in infants were found in the explored databases. The main characteristics of the articles included in this review are summarized in Table 1. The studied population was composed of 33 females and 29 males, with a mean age of 8762 days (IQ1 45 days—IQ3 105; IQR 60 days). Hemoptysis was the most prevalent symptom (52.45%), followed by epistaxis (8.2%) and hematemesis (3.27%). When onset symptoms were not reported, it meant that the infant was unresponsive. Respiratory distress with tachycardia and tachypnea was recorded in almost all cases. Endotracheal intubation showed the leakage of blood from the upper airways in 9.83% of cases. In only 11 out of 61 cases, exposure to risk factors was recorded [e.g., smoking tobacco; Stachybotrys chartarum, etc.]. 

#### 3.2.2. Instrumental Analysis and Laboratory Count (When Reported)

Laboratory analysis and instrumental analysis were recorded in almost all cases. Table 2 shows the different types of analyses that were performed. Chest X-rays were performed in all cases where instrumental data were reported. The most relevant finding was a bilateral diffuse infiltration with a radiological ground glass pattern [28]. This pattern was frequently mistaken for ongoing pneumonia and led physicians to administer antibiotics. In seven cases, CT was performed. Bilateral pulmonary involvement was reported just in two cases. Overall, 5 out of 61 cases presented leukocytosis, while anemia was recorded in only 8 cases. In agreement with the definition of AIPHI, when performed, the echocardiogram was negative. Further information is shown in Table 2. 

#### 3.2.3. Autopsy Data 

The screening of the updated literature revealed poor knowledge regarding idiopathic pulmonary hemorrhage in infants. Only two papers provided a complete description of the autoptic examination. No cases showed any signs of trauma or maltreatment. The literature evidence is shown in Table 3. 

In November 2007, Masoumi et al. [29] described the case of a 9-month-old female infant deceased from sudden infant death syndrome (SIDS). The baby was healthy and no other comorbidities were present. Familiarity with SIDS or other diseases was not reported. At the autopsy, the right and left lungs were heavy (weight 76 g and 68 g, respectively; expected range, 53–59 g), with a dark maroon surface and evidence of edema and congestion. The only peculiar findings were the presence of petechiae on the thymus, epicardium, and pleural surface. A histological examination was decisive and identified multiple foci of hemorrhage and intra-alveolar pulmonary siderophages in multiple random fields.

In this paper [29], blood infiltrates were evaluated using a semiquantitative measure of pulmonary intra-alveolar hemorrhage [absent: 0, severe: 4]. A random selection of 20 high-power fields (400×) was performed by the authors on four lung samples. The lung slide examinations revealed grade 3 intra-alveolar hemorrhages (PH), with a mean of 1732 pulmonary siderophages of 20 fields per specimen.

The siderophages were evaluated with iron staining on samples of both lungs, using the Prussian Blue method. At histological examination, 20 random fields were selected by the authors at 400× high-power fields. A total of 1732 hemosiderin and macrophages were found during the microscopy examination.

In another study, conducted by Hanzlick et al. (2001) [12], 60 cases of infant death were selected, regardless of the cause of death. Autopsies were performed in all cases, and lung specimens were collected in order to look for hemorrhages. For each case, four lung specimens were collected. For each microscopic evaluation of pulmonary hemorrhage, a semiquantitative 0–6 score [0: no hemorrhage; 6: diffuse hemorrhage] was assessed. In nine cases, there were extensive pulmonary hemorrhages, although, the cause of death was attributed to other pathologies in only five of them. In the other 4 cases classified as AIPHI, the score for lung hemorrhage was at least 10 [mean 15.6/24; mean score per HPF: 2.6]. One case showed hemosiderosis, with macrophages and sideral deposits in the form of hemosiderin. Based on these results, these cases of pulmonary hemorrhage in SIDS-diagnosed deceased infants could be classified as idiopathic acute pulmonary hemorrhage in infants.

## 4. Discussion

From the presented review of the updated literature regarding autoptic procedures on infants’ deaths, and from the autopsy record we presented, it is clear that idiopathic pulmonary hemorrhage is a rare but possible occurrence.

In general, pulmonary hemorrhage in infants is a medical emergency that requires proper diagnosis and treatment. The pathophysiological mechanism leads to blood extravasation within the respiratory spaces with or without evidence of capillaritis [30]. 

Pulmonary hemorrhage in newborns can have various causes with a nonspecific presentation, and this could become challenging for clinicians. Moreover, many other causes can mimic pulmonary hemorrhage (pulmonary abnormalities, etc.), and frequently, it can be confused with an infectious process or with SIDS.

Dyspnea and diffuse bilateral alveolar opacities are the most frequent presenting symptoms and signs of PH. In general, hemoptysis affects up to 66% of patients, and it may not be always present, even when hemorrhage is significant enough to cause severe anemia [31]. 

Foreign bodies, trauma, infections (e.g., bacterial, viral, fungal, and parasitic), cystic fibrosis, immunologic disease (e.g., Wegener’s granulomatosis, Goodpasture’s syndrome), neoplasms, pulmonary hemosiderosis, and congenital cardiovascular lesions have to be considered during differential diagnosis when hemoptysis presents in children [32].

Sometimes, hemoptysis has a detectable origin, as in the case of acute idiopathic pulmonary hemorrhage in infants (AIPHI). The Centers for Disease Control and Prevention defined AIPHI as “sudden onset of pulmonary hemorrhage in a previously healthy infant in whom differential diagnoses and neonatal medical problems that might cause pulmonary hemorrhage have been ruled out. Pulmonary hemorrhage can appear as hemoptysis or blood in the nose or airway with no evidence of upper respiratory or gastrointestinal bleeding. Patients have acute, severe respiratory distress or failure, requiring mechanical ventilation and chest radiograph (CXR), and usually demonstrate bilateral infiltrates”.

The attention on AIPHI increased when clusters of SIDSs, characterized by pulmonary hemorrhage, happened in Cleveland, Ohio, and Chicago in the 1990s. In these clusters, the newborns shared similar symptoms, such as respiratory distress and hemoptysis. 

Epidemiological studies led the scientific community to list the clinical criteria for the diagnosis of AIPHI and epidemiological controls of the phenomenon. They were:Previously healthy infant aged <1 year with a gestational age of >32 weeks.Abrupt or sudden onset of overt bleeding or obvious evidence of blood in the airway.Severe-appearing illness leading to acute respiratory distress or respiratory failure, resulting in hospitalization in a pediatric intensive care unit (PICU) or neonatal intensive care unit (NICU) with intubation and mechanical ventilation.Diffuse unilateral or bilateral pulmonary infiltrates visible on CXR or computerized tomography (CT) [33].

These criteria are useful to identify spontaneous hemorrhages of the lung in infants, but they cannot be considered exhaustive. As a matter of fact, in recent years, medical examiners are concerned about the arise of idiopathic pulmonary hemorrhage as a cause of unexpected infant death and that many cases may be misdiagnosed as SIDS. 

As shown in this review, hemoptysis is present in 52.45% of the cases, and it is followed by epistaxis (8.2%) and hematemesis (3.27%). Moreover, the clinical presentation of AIPHI was characterized by respiratory distress (tachypnea, tachycardia, dyspnea), pallor, anemia, and blood leakage from the endotracheal tube. Chest X-rays revealed unspecific signs that could be misdiagnosed with pneumonia. 

In Figure 3, an easy-to-read diagnostic flowchart is shown for cases of pulmonary hemorrhage with undefined causes.

We recorded a case of a 3-day-old boy that developed a pulmonary hemorrhage, whose symptoms began with acrocyanosis and polypnea, blood leakage from the endotracheal tube, anemia, respiratory failure, and hypotension.

No meconium aspiration signs were detected at the autoptic examination, nor were signs of asphyxia [3,34,35]. According to recent pathology reports [36] of PH, pulmonary hypoplasia is often associated with decreased pulmonary blood flow and vascular malformation, but their role is discussed. In our case, lung hypoplasia was investigated by the lung weight-to-body weight ratio (LW/BW) established by Wigglesworth et al. [37] and appeared to be less than two standard deviations. Blood discharge from the ET began immediately after intubation and no signs of acute emphysema related to barotrauma were found.

The hemorrhage remained sine causa, and no other causes of death could be supposed except AIPHI. In our case, pulmonary hypoplasia was not accompanied by the persistence of intra-acinar arterioles.

In this work, we suggest a criteriology for the assessment of post-mortem diagnosis of idiopathic pulmonary hemorrhage in infants. Based on the reviewed studies [12,29], and on the findings of the peculiar case we presented, the main pathological findings that can be found in AIPHI are: Increased weight of the lungs;Interstitial and endoalveolar hemorrhage with a diffuse or nodular pattern;Presence of endoalveolar hyaline membranes; (not necessary but possibly found)Accumulation of macrophages, siderophages, and hemosiderin as markers of previous bleeding; (not necessary, but possibly found);Absence of underlining conditions;

Pulmonary macrophages and siderophages in lung tissue may reflect previous episodes of pulmonary hemorrhage, so they must be interpreted beside the finding of extent of acute hemorrhage. As shown by our autopsy case, the onset period of symptoms can be variable and AIPHI can occur in the very first days of life, in the absence of other triggering conditions.

## 5. Conclusions

Acute idiopathic pulmonary hemorrhage is a quite rare disease characterized by a high mortality rate. It is frequently mistaken for SIDS. AIPHI needs to be well investigated. An epidemiological study conducted by the CDC in the 1990s suggested it to be related to genetic triggers. In this study, we present an autopsy record, showing a rare case of idiopathic pulmonary hemorrhage, which developed during the first days of life.

Therefore, we show how we managed the case and how we arrived at the post-mortem diagnosis. Thanks to literature research and our autoptic study [38], we propose pathological criteria to achieve the macroscopic and histological diagnosis of AIHPI. The novelty introduced by this study is that an autopsy record concerning a case of AIPHI in Europe has never been published in the world, the latest publication dates about 15 years ago. The presented findings could be helpful for forensic pathologists to achieve the post-mortem diagnosis of AIPHI, but further histological studies are necessary to standardize the criteria. 

## Figures and Tables

**Figure 1 diagnostics-13-01270-f001:**
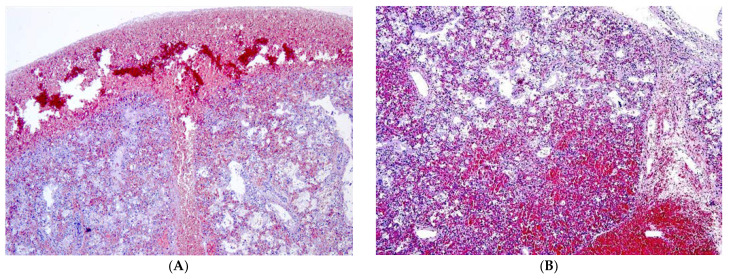
(**A**) The sample showed normal thickness of the pleura with red blood cells in the interstitium; (**B**) subpleural and alveolar hemorrhages. (**C**,**D**) The morphology of the pulmonary parenchyma appeared altered by septal and interalveolar hemorrhage with the presence of hyaline membranes. The arrows indicate hyaline membranes covering the alveolar lumen. (**A**: 4×; **B**: 2.5×; **C**: 20×; **D**: 20×).

**Figure 2 diagnostics-13-01270-f002:**
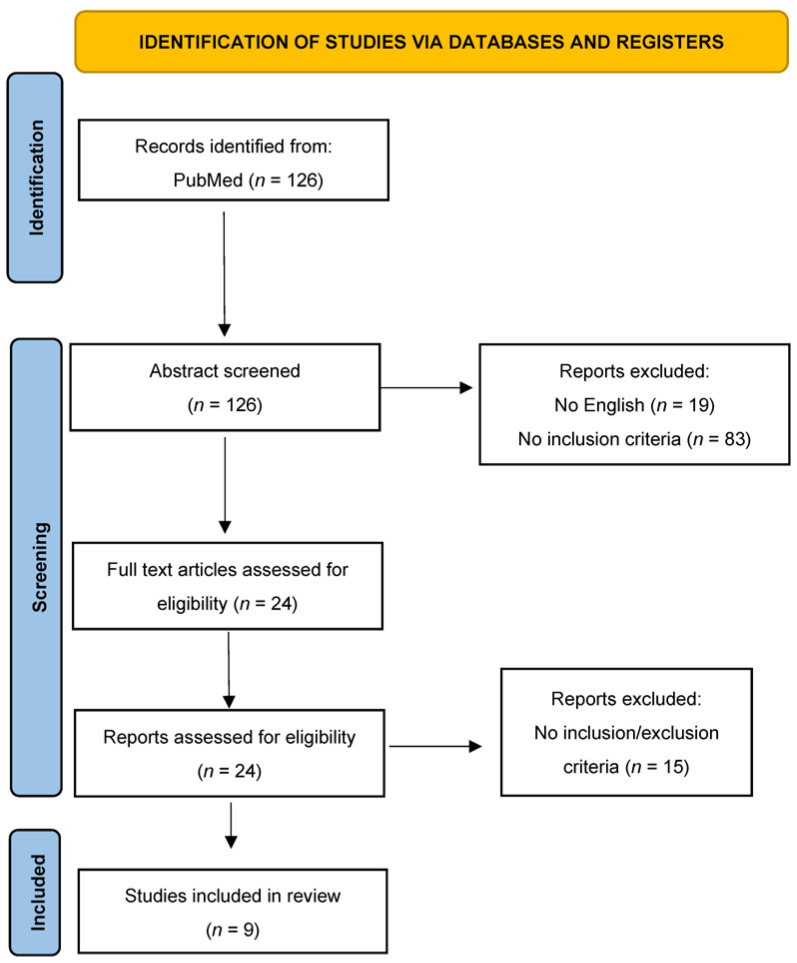
Search strategy: a methodological appraisal of each study was conducted according to the PRISMA standards, including an evaluation of bias. The data collection process included study selection and data extraction.

**Figure 3 diagnostics-13-01270-f003:**
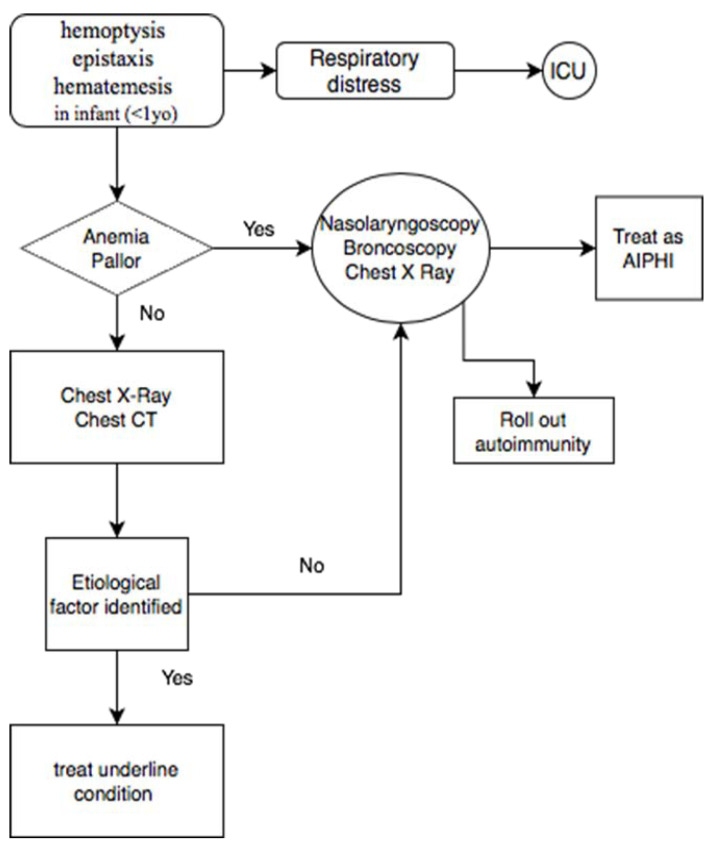
Diagnostic flowchart for patients presenting symptoms related to AIPHI.

**Table 1 diagnostics-13-01270-t001:** The table shows a summary of the findings of all reported clinical cases of acute idiopathic pulmonary hemorrhage in infants (AIPHI) in the literature. (M, male; F, female; N/A, not available).

n. (Patient)	Reference	Age	Sex	Vital Parameters and Condition at Hospital	Onset Symptoms	Exposure	Other
7	Chicago Cluster (1992–1994) [17]	3.0 (0.8–8) m	4/7 M	Respiratory distress	4/7 hemoptysis 2/7 epistaxis 1/7 blood leaking from the endotracheal tube	N/A	N/A
6	Pappas et al. (1996) [18]	Mean 2.3 m (0.9–6)	M	No fever, hypoxemia, respiratory acidosis	One previous infant seizure. 5/6 hemoptysis/hematemesis 1/6 epistaxis	N/A	Transfusion
37	Dearborn et al. (1999) [19,20,21] Cleveland Cluster	Mean 3.1 m	9 M 28 F	Tachypnea 30/37 respiratory distress	Acute onset with hemoptysis (18/24), lethargy, respiratory distress, apnea, bradycardia, seizures	10/37 tobacco smoke exposure 65% Stachybotrys chartarum in patient’s home	30/37 ventilator support 27/37 transfusion
1	Saeed et al. (1999) [22]	7.2 m	F	Not reported	Hemoptysis	N/A	Early prednisone
1	Chavez et al. (2000) [23]	27 d	M	Tachypneic, tachycardic (163 bpm), SpO_2_ 79%—O_2_ therapy with 10 l/min	Hemoptysis, blood leaking from the endotracheal tube	None	Early antibiotics
1	Novotny et al. (2000) [24]	40 d	M	Tachypnea (58/min) and respiratory distress SpO_2_ 76%, no fever, PaCO_2_ of 46 mm Hg, pH, 7.19, and PaO_2_, 74 mmHg on oxygen therapy	Blood leaking, suctioned from the mouth and posterior pharynx, subcostal retractions, pallor	Acute exposure to environmental tobacco smoke, fungal exposure, Penicillium—Trichoderma	Ampicillin and cefotaxime sodium
1	Al-Tamemi et al. (2009) [25]	34 w	M	Shallow breathing, SpO_2_ 84%, bilateral diffuse crackles, respiratory failure, severe metabolic acidosis, and low pCO_2_ due to hyperventilation	Unresponsive, face and clothing covered with blood	None	Broad-spectrum antibiotics
1	Gutierrez et al. (2014) [26]	5 w	M	Respiratory distress, tachycardic and tachypneic	Hematemesis	Not reported	Ceftriaxone Venovenous (VV) extracorporeal membrane oxygenation (ECMO)
4	Welsh et al. (2018) [27]	78 d	M	Respiratory distress	Hemoptysis	N/A	N/A
		32 d	M	N/A	Hemoptysis		
		36 d	M	Respiratory distress	N/A		
		38 d	M	Respiratory distress	Hemoptysis		
		1 m	F	Hemodynamic Shock	Epistaxis Blood leaking from the endotracheal tube	Not reported	Methylprednisolone surfactant antibiotics

**Table 2 diagnostics-13-01270-t002:** Data collection of laboratory and imaging findings of AIHPI from a systematic review of the current literature. (WBC, white blood cell; RBC, red blood cell; Hb, Hemoglobin; ANA, antinuclear antibody).

Reference	Laboratory Data	Instrumental Analysis	Other
Chicago Cluster (1992–1994) [17]	Not reported	Chest X-ray: bilateral infiltrates	Cultures of blood and urine specimens: negative for bacterial, mycotic, and viral pathogens Bronchoscopy: no source of bleeding
Pappas et al. (1996) [18]	Mean Hb 9.8 g/dL (range 7.3–14.7 g/dL) Platelets normal	Chest X-ray: Bilateral infiltrates Echocardiographic evaluation: normal myocardial contractility in all patients	Endotracheal aspirate for hemosiderin-laden macrophages: negative Serum cow’s milk precipitins: negative
Dearborn et al. (1999) [19,20,21] Cleveland Cluster	Not reported	Not reported	Bronchoscopy (22/37): hemosiderosis and chronic bleeding >6 months
Saeed et al. (1999) [22]	N/A	Chest X-ray: pulmonary infiltrates	N/A
Chavez et al. (2000) [23]	WBC 15.9 × 10^6^/mL (53% lymphocytes, 11% monocytes, 28% neutrophils) Platelet 256 × 10^6^/mL	Chest X-ray: bilateral hyperinflation, haziness in the upper lobe and lingula Normal lung perfusion scan	Immunoglobulin panel negative Complement panels negative Viral, bacterial, and fungal cultures negative ANA negative
Novotny et al. (2000) [24]	WBC 16 × 10^6^/mL Hb 12.3 g/dL Hematocrit 36% Platelet 624 × 10^6^/mL	Chest X-ray: diffuse bilateral alveolar infiltrates Skeletal survey: no trauma Echocardiogram: negative	Bronchial lavage fluid: hemosiderin and macrophages Antiglomerular basement membrane antibody: negative Antistreptolysin antibody level: normal
Al-Tamemi et al. (2009) [25]	Leukocytosis with WBC 22.8 × 10^6^/mL Lymphocytosis [×10^6^/mL] Neutrophil [×10^6^/mL] Platelets 555 × 10^6^/mL Hemoglobin 9.8 g/dL	Chest X-ray: bilateral ground glass appearance Computed tomography (CT) scan: bilateral alveolar opacities	Blood, sputum, urine, and stool cultures: negative Gastric aspirate: negative for hemosiderine
Gutierrez et al. (2014) [26]	Not reported	Chest X-ray: dense consolidation throughout the right lung and left lower lobe Echocardiogram: atrial fibrillation	N/A
Welsh et al. (2018) [27]	Not reported	4/4 Chest X-ray and CT not reported	3/4 flexible bronchoscopy: hemosiderin-laden macrophages
Sato et al. (2020) [28]	WBC 35.6 × 10^6^/mL Hb 11.8 g/dL	Chest X-ray: diffuse ground glass opacification of the left lung Echocardiogram: mild pulmonary hypertension but no congenital cardiac malformation Computed tomography (CT) scan: consolidation in the left upper and lower lobes	
	Worsening anemia	Chest X-ray: widespread ground glass opacification of the bilateral lungs Echocardiogram: negative Computed tomography (CT) scan: consolidation with air bronchogram in both lungs	Coagulation tests: normal

**Table 3 diagnostics-13-01270-t003:** Summary findings of the postmortem evaluation of AIPHI cases reported in the literature.

N. Cases	Reference	Age	Pulmonary Score (Mean)	Siderophages	Hemorrhage Localization (Lung)
1	Hanzlick et al. (2001) [12]	N/A	3	1/1	Alveolar and interstitial—Nodular
6	Masoumi et al. (2007) [29]	9 m	3	1/6	Alveolar—Nodular and diffuse

## Data Availability

Data sharing is not applicable; no new data were created or analyzed in this study.

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
