# Peer review of "Idiopathic Pulmonary Hemorrhage in Infancy: A Case Report and Literature Review"

_diagnostics, 2023, doi:10.3390/diagnostics13071270_

Round 1

Reviewer 1 Report

Dear Authors,

I have read with interest the manuscript ‘Idiopathic Pulmonary Hemorrhage in Infancy: a case report and Literature Review’.

Point 1: The writing is generally good and clear, but there are numerous English-language and grammar errors that indicate the article is written by a non-native speaker. The authors might wish to appoint a proofreader who can assist with this.

Few examples:

Line 3 : “that affects”

Lines 6-7: correct grammar

Lines 13-16: correct grammar

Line 24: “here it’s presented”

Line 30: “literature review” instead of “literally review”

In section 2.1 correct the following typos:

-Correct the acronym “APAGAR” in “APGAR”.

- the Authors might mean “Vaginal swab” instead of “vaginal Stab”

And so on…

Point 2: Section 2.3 should be renumbered in 3.1 after creating a new chapter named “3. Literature review”

Point 3: the bibliography in the introduction should be implemented with more recent literature. An example could be:

Barnes ME, Feeney E, Duncan A, Jassim S, MacNamara H, O'Hara J, Refila B, Allen J, McCollum D, Meehan J, Mullaly R, O'Cathain N, Roche E, Molloy EJ. Pulmonary haemorrhage in neonates: Systematic review of management. Acta Paediatr. 2022 Feb;111(2):236-244. doi: 10.1111/apa.16127. Epub 2021 Oct 28. PMID: 34582587.

Point 4: the Authors describe a case of acute pulmonary haemorrage and they assess an idiopathic diagnosis, based only on exclusion criteria, and they report a single case. The Authors suggest a criteriology of assessment of pathological diagnosis of idiopathic pulmonary hemorrhage in infants, based on the following criteria:
1. Increased weight of the lungs;

2. Interstitial and endoalveolar hemorrhage with diffuse or nodular pattern;

3. Presence of endoalveolar hyaline membranes; (possibile)

4. Accumulation of macrophages, siderophages and hemosiderin as markers of previous bleed-ing; (possibile);

5. Absence of underlining conditions

There are some issues that should be clarified before the case reported could be considered as idiopathic, even according the same purpose of the Authors.

Firstly, the condition of “absence of underlining conditions” seems not adequately investigated.

The child showed signs of fetal suffering, and the Authors report the presence of bradycardia at CTG the mild bradycardia, with the execution of an emergency cesarean section. Moreover the Authors wrote “The airways show hyperemia of the mucous membrane of the larynx and trachea, which are covered with crimson-colored liquid.”

Have you performed a microbiological testing for the research of infections? How do interpret the finding of pleural thickening?

A recent study of a retrospective autopsy case series about perinatal pulmonary hemorrhages, suggested a possible explanation in the presence of either maternal or fetal vascular malperfusion, as well as  Pulmonary hypoplasia and/or persistence of intra-acinar arterioles.

Agarwal I, Ernst LM. Perinatal Pulmonary Hemorrhage: A Retrospective Autopsy Case Series. Pediatr Dev Pathol. 2020 Aug;23(4):267-273. doi: 10.1177/1093526619900728. Epub 2020 Feb 19. PMID: 32075513.

We have to take into account that the consideration that the weight of the lungs was inferior to the mean expected lung weight.
At this regard, I suggest the following reference that considers “Pulmonary Hypolasia” as a lung weight more than 2 standard deviations below the normal for age (or gestational age), or in terms of lung weight–to–body weight ratio, the normal being 0.222±0.002 for term and near-term infants.

Jean-Martin Laberge, Pramod Puligandla, Chapter 64 - Congenital Malformations of the Lungs and Airways, Editor(s): Lynn M. Taussig, Louis I. Landau, Pediatric Respiratory Medicine (Second Edition), Mosby, 2008, Pages 907-941, ISBN 9780323040488, https://doi.org/10.1016/B978-032304048-8.50068-2.) (https://www.sciencedirect.com/science/article/pii/B9780323040488500682)

Have you investigated the intra-acinar vascular structure? Have you investigated the placenta, in consideration of the fetal suffering that led to a C-Section?

The images of the histological specimens show a tendency to atelectasis. The presence of blood in the airways can be also due to an intubation procedure. Were there signs for it? Have you investigated a lack of surfactant or the hypothesis of a barotrauma? Have you got more images in order to evaluate the septal alveolar integrity?

Moreover the presence of hyaline membranes is usually a common finding suggesting for a distress respiratory Syndrome due to a lack of surfactant. This condition can occur also because of the mecoal aspiration. How do you exclude that this could be due to a malperfusion or disventilation?

Point 5: the references should follow the style of the journal, for example [1-3] instead of [1]-[3].

Author Response

Reviewer 1

We would like to thank the Reviewer for suggestions and for giving us the opportunity to submit a revised version of our manuscript.

 We appreciate the Reviewer’s insight and we hope that we accommodated the suggestions in the revised manuscript. You can find all changes in the revised manuscript in yellow.

Question

Dear Authors, I have read with interest the manuscript ‘Idiopathic Pulmonary Hemorrhage in Infancy: a case report and Literature Review’. Point 1: The writing is generally good and clear, but there are numerous English-language and grammar errors that indicate the article is written by a non-native speaker. The authors might wish to appoint a proofreader who can assist with this. Few examples: Line 3 : “that affects” Lines 6-7: correct grammar Lines 13-16: correct grammar Line 24: “here it’s presented” Line 30: “literature review” instead of “literally review” In section 2.1  correct the following typos: -Correct the acronym “APAGAR” in “APGAR”. - the Authors might mean “Vaginal swab” instead of “vaginal Stab” And so on…”

Answer

Dear reviewer thanks so much for your comment.

As you advised, we decided on a comprehensive grammar and syntactic revision of the English and corrected typos. You can find all changes in the text.

Question:

Point 2: Section 2.3 should be renumbered in 3.1 after creating a new chapter named “3. Literature review

Answer

We created a new section, as you advised: “3. Literature review - 3.1. Materials and Methods – 3.2 Results - 3.2.1. General presentation and clinical manifestation of AIPHI…”.

Question 

Point 3: the bibliography in the introduction should be implemented with more recent literature. An example could be:

Barnes ME, Feeney E, Duncan A, Jassim S, MacNamara H, O'Hara J, Refila B, Allen J, McCollum D, Meehan J, Mullaly R, O'Cathain N, Roche E, Molloy EJ. Pulmonary hemorrhage in neonates: Systematic review of management. Acta Paediatr. 2022 Feb;111(2):236-244. doi: 10.1111/apa.16127. Epub 2021 Oct 28. PMID: 34582587.

 Answer

We changed the introduction following your suggestion by including the reference [2].

Question 

“Point 4: The Authors describe a case of acute pulmonary hemorrhage and they assess an idiopathic diagnosis, based only on exclusion criteria, and they report a single case. The Authors suggest a criteriology of assessment of pathological diagnosis of idiopathic pulmonary hemorrhage in infants, based on the following criteria… There are some issues that should be clarified before the case reported could be considered as idiopathic, even according the same purpose of the Authors.

Firstly, the condition of “absence of underlining conditions” seems not adequately investigated.

The child showed signs of fetal suffering, and the Authors report the presence of bradycardia at CTG the mild bradycardia, with the execution of an emergency cesarean section. Moreover, the Authors wrote “The airways show hyperemia of the mucous membrane of the larynx and trachea, which are covered with crimson-colored liquid.” Have you performed a microbiological testing for the research of infections? How do interpret the finding of pleural thickening?”

Answer

Thank you so much for your comment. The “crimson-colored liquid” found at the autopsy in the airways, seems to be linked to both intubation and blood leakage. As far as we know about clinical history, in life microbiological tests have not been performed. Also, we did not perform postmortem microbiological analysis because there was no clinical and histological evidence of infection.

 Infection disease was excluded by autopsy, and also no infectious foci were found at histological analysis. According to Bendapudi P. et al. (https://doi.org/10.1016/j.paed.2012.08.008), sepsis is a recognized cause of pulmonary haemorrhage (PH) but, in our case, there was no clinical evidence of infectious signs (leukocytosis, RPC, PCT etc.…). We hope to better clarify the absence of perinatal sepsis by inserting: “Multiple blood tests were performed during hospitalization and no evidence of leukocytosis or high inflammatory indices was detected.”.

The pleura thickening was an isolated finding due to blood infiltration in the interstitium, without macroscopic evidence of infection. Pleural mesothelium was normal despite fetal development of visceral pleura proceeding asynchronously [see Ann Anat (1996) 178: 91- 99 - is normal to find different stages of pleural mesothelium stratification].

Question

“A recent study of a retrospective autopsy case series about perinatal pulmonary hemorrhages, suggested a possible explanation in the presence of either maternal or fetal vascular malperfusion, as well as Pulmonary hypoplasia and/or persistence of intra-acinar arterioles.

Agarwal I, Ernst LM. Perinatal Pulmonary Hemorrhage: A Retrospective Autopsy Case Series. Pediatr Dev Pathol. 2020 Aug;23(4):267-273. doi: 10.1177/1093526619900728. Epub 2020 Feb 19. PMID: 32075513.

We have to take into account that the consideration that the weight of the lungs was inferior to the mean expected lung weight. At this regard, I suggest the following reference that considers “Pulmonary Hypolasia” as a lung weight more than 2 standard deviations below the normal for age (or gestational age), or in terms of lung weight–to–body weight ratio, the normal being 0.222±0.002 for term and near-term infants.  

Jean-Martin Laberge, Pramod Puligandla, Chapter 64 - Congenital Malformations of the Lungs and Airways, Editor(s): Lynn M. Taussig, Louis I. Landau, Pediatric Respiratory Medicine (Second Edition), Mosby, 2008, Pages 907-941, ISBN 9780323040488.)

(https://www.sciencedirect.com/science/article/pii/B9780323040488500682)

Have you investigated the intra-acinar vascular structure?

Answer

Thanks for this important insight. Indeed, the lung weight–to–body weight ratio is lower than the standard (0.195), but we don’t know the mean weight of newborn lungs in the region where the childbelongs. Otherwise, Laberge J.M. et al. described “The patient is tachypneic, with restricted chest wall movement, and in respiratory distress. Less severe degrees of hypoplasia—unilateral or bilateral—may present later with persistent tachypnea or disproportionate shortness of breath with exercise.”. In accordance with Agarwal I et Al. lung hypoplasia is found only in 41.17% cases of PH, and his role in etiopathogenesis of PH is discussed.

In our study, we excluded any of the most common causes of pulmonary hypoplasia [oligohydramnios, space occupying lesions, thoracic cage anomalies and conditions preventing normal fetal breathing movements]. In the case, we reported a sudden development of cyanosis and tachypnea due to devasting lung hemorrhage. Furthermore, tachypnea and cyanosis are a manifestation of both diseases, but AIPHI is the only relatable cause once pulmonary hypoplasia was excluded. We followed your suggestion and we revised all the histological samples to investigate intra-acinar arterioles persistence. No vascular malformations were founded.

To improve the manuscript with your essential contribution, we implement the manuscript with:

Section 2.2: “A lung weight to body weight ratio (LW/BW) of 0.0195 revealed pulmonary hypoplasia.”

Section 2.3: “No signs of abnormal pulmonary angiogenesis were founded.”

Section 4: “According to recent pathology reports [10.1177/1093526619900728] of PH, pulmonary hypoplasia is often associated with decreased pulmonary blood flow and vascular malformation. Lung hypoplasia evaluation was investigated by lung weight to body weight ratio (LW/BW) established by Wigglesworth et al. and was less than two standard deviations. In our case, pulmonary hypoplasia was not accompanied by the persistence of intra-acinar arterioles”.

Question

Have you investigated the placenta, in consideration of the fetal suffering that led to a C-Section?

Answer

Thanks for your suggestion. Placenta investigation is mandatory for theglobal detection of sudden infant death. Placental weight is unknown but from a revision of histological samples we see no placental malformation. We add: Section 2.3 “histological examination of the placenta [350 gr] revealed no malformations.”

Question

The images of the histological specimens show a tendency to atelectasis. The presence of blood in the airways can be also due to an intubation procedure. Were there signs for it? Have you investigated a lack of surfactant or the hypothesis of a barotrauma? Have you got more images in order to evaluate the septal alveolar integrity? Moreover, the presence of hyaline membranes is usually a common finding suggesting for a distress respiratory Syndrome due to a lack of surfactant. This condition can occur also because of the mecoal aspiration. How do you exclude that this could be due to a malperfusion or disventilation?

Answer

To better investigate lung hemorrhage, we provide other images that show the same features. Moreover, we didn’t find real atelectasis, only patchy areas of incomplete collapsed alveolar spaces. This is a common funding in acute respiratory distress syndrome (ARDS). The presence of hyaline membranes is frequently found in ARDS also known as hyaline membrane disease. Indeed, we suppose that the first cause was the lack of surfactant but alveolar spaces were opened and clear. This is coherent with the gestational period and the age of the newborn.

We better clarify this point as: “section 2.3 Minimal areas of partial alveolar collapse were found but there was no presence of evident atelectasis”.

Moreover, respiratory rate/oxygen saturation was normal for a 1 day old boy. Meconium aspiration was clearly excluded for two reasons; first of all, aspiration of upper airways was performed by the obstetrician, and no mconium fluid was found at the histological examination. We added in the manuscript: “section 4: No meconium aspiration signs were detected at the autoptic examination…”.

After an evaluation of the clinical records, we excluded the hypothesis of barotrauma because blood discharge began after ET insertion. There were no areas of acute septal ruptures that justify this acute and dramatic hemorrhage: “section 4: … Blood discharge from the ET began immediately after intubation and no signs of acute emphysema related to barotrauma were found.”.

Question

“Point 5: the references should follow the style of the journal, for example [1-3] instead of [1]-[3].”

Answer

Thank for this comment, we modified as you suggested.

Reviewer 2 Report

Very interesting and original study

Author Response

Reviewer 2

Dear reviewer,

We made an extensive English revision. [highlighted - light blue]

Thank you so much

Sincerely

Reviewer 3 Report

"In general, pulmonary haemorrhage in infants s a medical emergency requiring expedient diagnosis and treatment." 

I'll change in: 

"In general, pulmonary haemorrhage in infants Is a medical emergency THAT REQUIRES PROPER DIAGNOSIS AND TREATMENT."

I would say something more about the criteria.

Author Response

Reviewer 3

Dear reviewer,

Thank you so much. We made an extensive English revision.

We changed the sentence following your suggestion: “section 4: In general, pulmonary hemorrhage in infants is a medical emergency that requires proper diagnosis and treatment.” [highlighted in green]. We better specified criteria according to the rates of presentation and the relevance.

Sincerely

Round 2

Reviewer 1 Report

Dear Authors, thanks for solving the issues.

It can be a good peace of scholarship.